# Extracellular HSP90α Induces MyD88-IRAK Complex-Associated IKKα/β−NF-κB/IRF3 and JAK2/TYK2−STAT-3 Signaling in Macrophages for Tumor-Promoting M2-Polarization

**DOI:** 10.3390/cells11020229

**Published:** 2022-01-11

**Authors:** Chi-Shuan Fan, Chia-Chi Chen, Li-Li Chen, Kee Voon Chua, Hui-Chen Hung, John T. -A. Hsu, Tze-Sing Huang

**Affiliations:** 1National Institute of Cancer Research, National Health Research Institutes, Miaoli 350, Taiwan; change0935693367@hotmail.com (C.-S.F.); poseking2430@nhri.org.tw (C.-C.C.); lilichen@nhri.org.tw (L.-L.C.); priscillachua74@gmail.com (K.V.C.); 2Institute of Biotechnology and Pharmaceutical Research, National Health Research Institutes, Miaoli 350, Taiwan; huichen@nhri.edu.tw (H.-C.H.); tsuanhsu@nhri.edu.tw (J.T.-A.H.); 3Department of Biochemistry, School of Medicine, Kaohsiung Medical University, Kaohsiung 807, Taiwan; 4Ph. D. Program in Tissue Engineering and Regenerative Medicine, Biotechnology Center, National Chung Hsing University, Taichung 402, Taiwan

**Keywords:** tumor-infiltrating macrophage, M2-polarization, extracellular HSP90α (eHSP90α), CD91, TLR4

## Abstract

M2-polarization and the tumoricidal to tumor-promoting transition are commonly observed with tumor-infiltrating macrophages after interplay with cancer cells or/and other stroma cells. Our previous study indicated that macrophage M2-polarization can be induced by extracellular HSP90α (eHSP90α) secreted from endothelial-to-mesenchymal transition-derived cancer-associated fibroblasts. To extend the finding, we herein validated that eHSP90α-induced M2-polarized macrophages exhibited a tumor-promoting activity and the promoted tumor tissues had significant increases in microvascular density but decreases in CD4^+^ T-cell level. We further investigated the signaling pathways occurring in eHSP90α-stimulated macrophages. When macrophages were exposed to eHSP90α, CD91 and toll-like receptor 4 (TLR4) functioned as the receptor/co-receptor for eHSP90α binding to recruit interleukin (IL)-1 receptor-associated kinases (IRAKs) and myeloid differentiation factor 88 (MyD88), and next elicited a canonical CD91/MyD88–IRAK1/4–IκB kinase α/β (IKKα/β)–nuclear factor-κB (NF-κB)/interferon regulatory factor 3 (IRF3) signaling pathway. Despite TLR4-MyD88 complex-associated activations of IKKα/β, NF-κB and IRF3 being well-known as involved in macrophage M1-activation, our results demonstrated that the CD91-TLR4-MyD88 complex-associated IRAK1/4−IKKα/β−NF-κB/IRF3 pathway was not only directly involved in M2-associated CD163, CD204, and IL-10 gene expressions but also required for downregulation of M1 inflammatory cytokines. Additionally, Janus kinase 2 (JAK2) and tyrosine kinase 2 (TYK2) were recruited onto MyD88 to induce the phosphorylation and activation of the transcription factor signal transducer and activator of transcription-3 (STAT-3). The JAK2/TYK2−STAT-3 signaling is known to associate with tumor promotion. In this study, the MyD88−JAK2/TYK2−STAT-3 pathway was demonstrated to contribute to eHSP90α-induced macrophage M2-polarization by regulating the expressions of M1- and M2-related genes, proangiogenic protein vascular endothelial growth factor, and phagocytosis-interfering factor Sec22b.

## 1. Introduction

Myeloid-derived cells are commonly found in tumor tissues and are associated with cancer development and progression [1]. Macrophages are one of the most abundant tumor-infiltrating myeloid-derived cells, and their infiltration levels have been clinically correlated with poor prognosis in many malignancies including invasive breast carcinoma [2], non-small cell lung cancer [3], and pancreatic ductal adenocarcinoma (PDAC) [4]. Tumor-infiltrating macrophages (TIMs) were initially thought to be associated with inflammatory reactions and exhibited innate tumoricidal function [5]. However, accumulating evidence has demonstrated that TIMs develop a reverse role after their interactions with tumor cells, stromal cells, and factors produced within the tumor microenvironment, and promote tumor angiogenesis, growth, spreading, and immunosuppression [6,7,8,9]. This anti- to pro-tumor transition is generally associated with M2-polarization, which is characterized by the production of anti-inflammatory factors such as interleukin (IL)-10 and transforming growth factor-β (TGF-β). The M2-polarization, or so-called “alternative activation”, of macrophages is so named to distinguish it from the M1 type, which is classically activated by bacterial cell-wall components such as lipopolysaccharides (LPS), intracellular pathogens, or inflammatory cytokines such as interferon-γ (IFN-γ) and tumor necrosis factor-α (TNF-α) [10]. The already known M2 activators include fungi, parasites, apoptotic tissue cells, immune complexes, TGF-β, glucocorticoid, macrophage colony-stimulating factor (M-CSF), and Th2 cytokines IL-4, IL-10, and IL-13 [10]. Many M2 markers have been defined based on the early investigation of the macrophages treated with IL-4 and IL-13 [11], but not all of these markers can be detected in macrophages when stimulated by other M2 activators. Several M2 subtypes have been proposed to reflect the idea that macrophages can exhibit their high plasticity and tune their phenotypes in response to different stimuli [10].

The M2-polarization of TIMs can be facilitated by other stromal cells [8]. Cancer-associated fibroblasts (CAFs) are a predominant population of cancer stromal cells. A significant correlation between the levels of M2-macrophages and CAFs has been revealed in several malignancies such as oral squamous cell carcinoma [12], colorectal cancer [13], and PDAC [14]. The factors such as IL-6, IL-8, stromal-derived growth factor-1 (SDF-1), IL-33, IL-10, and TGF-β can be secreted from CAFs to increase the recruitment and M2-polarization of macrophages [15,16,17,18]. Our previous study has also indicated that HSP90α secreted from endothelial-to-mesenchymal transition (EndoMT)-derived CAFs is able to induce macrophage M2-polarization [14]. The secreted extracellular HSP90α (eHSP90α) does not only induce several M2 markers such as CD163, CD204, IL-10, and TGF-β, but also elicits a feedforward loop leading to expression and secretion of large amount of HSP90α from the M2-polarized macrophages [14]. HSP90α was originally recognized as a chaperone that aids in maturation, trafficking, and activities of client proteins [19]. It can also be secreted from keratinocytes, fibroblasts, and cancer cells, functioning from regulation of inflammatory reactions in wound healing to promotion of cancer cell malignancy through inducing tumor cell epithelial-mesenchymal transition (EMT), migration, invasion, stemness, and metastasis [20,21,22].

Clinically, elevated levels of eHSP90α have been detected from the serum specimens of the patients diagnosed with pancreatitis or early- or late-staged PDAC [23]. In PDAC-developing LSL-K-Ras^G12D^/Pdx1-Cre transgenic mice, an evident increase of serum HSP90α levels has been detected in 3-month-old mice which persists at 6 months of age [23]. These mutant K-Ras-harboring mice develop PDAC through a stage-by-stage process: an acinar-to-ductal metaplasia (ADM) stage occurring within 3 months after birth, followed by a pancreatic intraepithelial neoplasia (PanIN) stage from 3 to 6 months after birth, and the final formation of PDAC lesions after 6 months of age. Their PDAC development can be effectively inhibited when they are administered an eHSP90α inhibitor 1 month after birth [23]. Interestingly, pancreas-infiltrating myeloid-derived macrophages have also been detected as early as the ADM stage and are required for the ADM process and elevation of serum HSP90α levels [23]. These macrophages secrete not only significant amounts of HSP90α but also IL-6 and IL-8 to stimulate a Janus kinase 2 (JAK2)/tyrosine kinase 2 (TYK2)-signal transducer and activator of transcription-3 (STAT-3) signaling pathway in pancreatic ductal epithelial cells to express and secrete more HSP90α [23]. Moreover, eHSP90α per se also stimulates significant HSP90α expression and secretion from macrophages and epithelial cells [14,24]. In the resultant eHSP90α-rich tissue microenvironment, eHSP90α would not only be a potent inducer of HSP90α-secreting M2-type macrophages, but also acts as a promoter of EMT, migration, and invasion of pancreatic ductal epithelial cells [14,23,24]. Besides functioning in malignant progression, TIMs and the associated eHSP90α are also critically involved in the development of PDAC.

Considering M2-polarized TIMs are closely associated with cancer initiation and progression, identification of the molecules and pathways driving the M2 polarization will provide clues to develop M2-focused cancer preventive and therapeutic strategies. Like TGF-β, eHSP90α has both “afferent” and “efferent” roles with regard to macrophage M2-polarization. However, how eHSP90α acts on macrophages is not well known. Our previous study suggested that cell-surface CD91 and toll-like receptor 4 (TLR4) can function as the receptor/co-receptor for eHSP90α binding on macrophages [14]. The eHSP90α binding enhances CD91−TLR4 association and further recruitment of myeloid differentiation factor 88 (MyD88). JAK2 and TYK2 are then recruited onto MyD88 to turn on a MyD88−JAK2/TYK2−STAT-3 signaling pathway to induce more HSP90α expression and secretion [14]. It is intriguing and remains to be investigated whether this signaling pathway is also involved in other events for macrophage M2-polarization. Besides JAKs, IL-1 receptor-associated kinases (IRAKs) are known to associate with MyD88 to trigger signaling cascades resulting in the activations of IκB kinases (IKKs) and the downstream transcriptional factors nuclear factor-κB (NF-κB) and IFN regulatory factor 3 (IRF3) [25,26,27]. It is intrinsic and important to clarify whether the MyD88−IRAKs−IKKs−NF-κB/IRF3 signaling pathway is also responsible for eHSP90α-induced macrophage M2-polarization.

In this study, we first validated the tumor-promoting activity exhibited by eHSP90α-induced M2-type macrophages and further investigated the signaling pathways occurring in eHSP90α-treated macrophages. Our results revealed that, through the macrophage cell-surface receptor CD91 and TLR4, eHSP90α induced a MyD88−IRAK1/4−IKKα/β−NF-κB/IRF3 pathway for downregulation of M1-associated IL-1β and TNF-α levels and upregulation of M2-associated TGF-β, CD163, CD204, and IL-10 expressions. Furthermore, a CD91 and TLR4 receptor complex-associated MyD88−JAK2/TYK2−STAT-3 pathway was also induced by eHSP90α to regulate the expressions of M1- and M2-markers, proangiogenic factor vascular endothelial growth factor (VEGF), and phagocytosis-interfering factor Sec22b.

## 2. Materials and Methods

### 2.1. Cell Cultures

All cell cultures were performed in 37 °C and 5% CO_2_ humidified incubators. Human monocytic leukemia THP-1 cells and mouse endothelial cell line 3B-11 (ATCC CRL-2160^TM^, American Type Culture Collection, Manassas, VA, USA) were cultivated in RPMI-1640 medium supplemented with 10% fetal bovine serum (FBS) and a mix of 100 units/mL of penicillin, 100 μg/mL of streptomycin, and 2 mM of _L_-glutamine (1× PSG). Mouse macrophage line RAW264.7 was cultured with Dulbecco’s Modified Eagle’s Medium (DMEM) containing 10% FBS and 1× PSG. Human pancreatic ductal epithelial cell line HPDE was cultivated in keratinocyte serum-free (KSF) medium supplemented with bovine pituitary extract and epidermal growth factor (Life Technologies, Grand Island, NY, USA). Human umbilical vein endothelial cells (HUVECs) were isolated as described previously and subcultured with M199 medium plus 20% of FBS, 30 μg/mL of endothelial cell growth supplement (EMD Millipore, Billerica, MA, USA), 100 units/mL of penicillin, and 100 μg/mL of streptomycin [14]. For preparation of mouse bone-marrow-derived macrophages (BMDMs), bone marrow cells were isolated from C57BL/6 mice and incubated 7 days with DMEM plus 10% FBS, 20% L929-conditioned medium, and 1× PSG. Adherent cells were collected as BMDMs and maintained in DMEM plus 10% FBS and 1× PSG. To study the effects of eHSP90α on macrophages, THP-1-derived macrophages, RAW264.7 cells, and BMDMs were preincubated with 0.5% serum-containing medium for 16 h before the further treatment with phosphate-buffered saline (PBS) or 15 μg/mL of purified recombinant HSP90α (rHSP90α; Enzo Life Sciences Inc., Farmingdale, NY, USA).

### 2.2. Preparation of Conditioned Media (CM)

To induce differentiation to macrophages, THP-1 cells were treated for 24 h with 100 ng/mL of 12-*O*-tetradecanoyl-13-phorbol acetate. Adherent cells were harvested as macrophages and seeded onto 10 cm dishes at a density of 1 × 10^6^ cells per dish in 10 mL serum-free RPMI-1640 medium. A dish containing 10 mL serum-free medium without macrophages was set up in parallel as a control. After incubation for 24 h, the macrophage-conditioned and control media were collected, centrifuged, and filtered through 0.45 μm filters, and designated as “Mϕ-CM” and “Ctrl”, respectively. To prepare HPDE-conditioned media, 2 × 10^6^ HPDE cells were seeded onto each 10 cm dish and incubated with Ctrl or Mϕ-CM for 24 h. After being washed twice with PBS, treated HPDE cells were incubated with 10 mL of fresh culture medium for another 24 h. The media were collected, centrifuged, and filtered through 0.45 μm filters, and designated as “HPDE(Ctrl)-CM” and “HPDE(Mϕ-CM)-CM”, respectively. To prepare the CM of endothelial and EndoMT-derived cells, HUVECs and 3B-11 cells (2 × 10^6^ cells per 10 cm dish) were treated for 24 h with PBS or 0.3 μg/mL of osteopontin in their respective low-serum culture media as described previously [14]. After being washed twice with PBS, the endothelial and EndoMT-derived cells were incubated with 10 mL of fresh low-serum media for another 24 h. Dishes of the respective low-serum culture media without HUVECs and 3B-11 cells were set up simultaneously as control media. The media were collected, centrifuged, and filtered with 0.45 μm filters, and designated as “Ctrl”, “Endo-CM”, and “EndoMT-CM”, respectively.

### 2.3. RNA Isolation and Quantitative RT-PCR

Total RNA of treated macrophages was isolated using TRIzol reagent (Thermo Fisher Scientific, Waltham, MA, USA). Furthermore, RNA was converted to cDNA by Tetro Reverse Transcriptase (Bioline Reagents Ltd., London, UK). The cDNA products were used as the templates for PCR analyses. The primers and PCR conditions were listed in Appendix A. Quantitative RT-PCR (qPCR) was performed with QuantiNova SYBR Green RT-PCR Kit (Qiagen, Hilden, Germany) in StepOnePlus^TM^ Real-Time PCR System (Thermo Fisher Scientific).

### 2.4. Phagocytosis Assay

Macrophages seeded onto glass coverslips at a density of 1 × 10^5^ cells per 22 × 22 mm coverslip were incubated 16 h with 0.5% serum-containing medium and then treated as indicated in the figure legends. The treated macrophages were further incubated with 0.02% (*w*/*v*) of FBS-opsonized latex beads (Sigma-Aldrich, St. Louis, MO, USA) for another 3 h. After being washed trice with PBS and fixed with 1% paraformaldehyde, the macrophage nuclei were stained with 4′,6′-diamidino-2-phenylindole (DAPI; Sigma-Aldrich). Finally, the images were visualized and photographed using Leica TCS SP5 II confocal microscope and LAS AF Lite 4.0 software (Leica, Wetzlar, Germany).

### 2.5. Proximity Ligation Assay (PLA)

Macrophages seeded onto glass coverslips at a density of 2 × 10^5^ cells per 22 × 22 mm coverslip were preincubated 16 h with 0.5% serum-containing medium and treated with PBS or 15 μg/mL of rHSP90α for another 4 h. For assaying the associations of eHSP90α with CD91 and TLR4, PBS or rHSP90α-treated macrophages were fixed with 3% paraformaldehyde and then blocked with the blocking solution supplied in the Duolink in situ PLA kit (Sigma-Aldrich). For assaying CD91−MyD88, TLR4−MyD88, CD91−IRAK1, MyD88−IRAK1, MyD88−JAK2, and MyD88−TYK2 associations, PBS or rHSP90α-treated macrophages were fixed with 3% paraformaldehyde, permeabilized with 0.1% Triton X-100, and then blocked with the blocking solution. These treated macrophage samples were further incubated overnight at 4 °C with the combinations of two primary antibodies: rabbit anti-HSP90α (1:80, cat. #AHP-1339, AbD Serotec, Raleigh, NC, USA) plus goat anti-TLR4 (1:40, cat. #sc-8694, Santa Cruz Biotechnology, Santa Cruz, CA, USA), rabbit anti-HSP90α (1:80, cat. #AHP-1339, AbD Serotec) plus mouse anti-CD91 (1:80, cat. #550495, BD Biosciences, San Jose, CA, USA), goat anti-TLR4 (1:40, cat. #sc-8694, Santa Cruz Biotechnology) plus rabbit anti-MyD88 (1:100, cat. #ab2068, Abcam, Cambridge, UK), mouse anti-CD91 (1:80, cat. #550495, BD Biosciences) plus rabbit anti-MyD88 (1:100, cat. #ab2068, Abcam), mouse anti-CD91 (1:80, cat. #550495, BD Biosciences) plus rabbit anti-IRAK1 (1:80, cat. #4504, Cell Signaling, Danvers, MA, USA), rabbit anti-MyD88 (1:100, cat. #ab2068, Abcam) plus mouse anti-IRAK1 (1:40, cat. #sc-5288, Santa Cruz Biotechnology), rabbit anti-MyD88 (1:100, cat. #ab2068, Abcam) plus goat anti-JAK2 (1:40, cat. #sc-34480, Santa Cruz Biotechnology), and rabbit anti-MyD88 (1:100, cat. #ab2068, Abcam) plus goat anti-TYK2 (1:40, cat. #sc-30671, Santa Cruz Biotechnology). After being washed with Tris-buffered saline plus 0.05% Tween-20, the macrophage samples were subjected to subsequent agents and procedure fully according to the manufacturer’s instructions of the Duolink in situ PLA kit (Sigma-Aldrich). Finally, macrophage nuclei were counterstained with DAPI, and the images were photographed and analyzed using Leica TCS SP5 II confocal microscope and LAS AF Lite 4.0 software (Leica).

### 2.6. Immunoblot Analysis

Cell lysates of the macrophages treated as indicated in figures were prepared by briefly sonicating cells in lysis buffer [23] supplemented with cocktails of protease inhibitors and phosphatase inhibitors (Sigma-Aldrich). SDS-polyacrylamide gel electrophoreses and immunoblot analyses were performed according to the general procedures. The immunoreactive protein bands were detected by enhanced chemiluminescence (Luminata^TM^ Crescendo Western HRP Substrate, EMD Millipore). The band intensities were quantified using ImageJ software (National Institutes of Health, Bethesda, MD, USA). The primary antibodies used were specific for p-IRAK1 (1:500, cat. #sc-130197, Santa Cruz Biotechnology), IRAK1 (1:500, cat. #sc-5288, Santa Cruz Biotechnology), p-IRAK4 (1:1000, cat. #11927, Cell Signaling), IRAK4 (1:500, Cat. #sc-374349, Santa Cruz Biotechnology), p-IKKα/β (1:1000, cat. #2697, Cell Signaling), IKKα (1:500, cat. #3285, Epitomics, Burlingame, CA, USA), IKKβ (1:500, cat. #sc-7329, Santa Cruz Biotechnology), p-NF-κB (1:1000, cat. #1546-1, Epitomics), NF-κB (1:5000, cat. #sc-516102), p-IRF3 (1:1000, cat. #4947, Cell Signaling), IRF3 (1:500, cat. #sc-33641, Santa Cruz Biotechnology), p-JAK2 (1:1000, cat. #3771, Cell Signaling), JAK2 (1:1000, cat. #3230, Cell Signaling), p-TYK2 (1:1000, cat. #9312, Cell Signaling), TYK2 (1:1000, cat. #9321, Cell Signaling), p-STAT-3 (1:500, cat. #2236, Epitomics), STAT-3 (1:1000, cat. #04-1014, EMD Millipore), and GAPDH (1:20,000, cat. #NB300-221, Novus Biologicals, Littleton, CO, USA).

### 2.7. Enzyme-Linked Immunosorbent Assay (ELISA)

Secretion levels of IL-1β, IL-10, and TGF-β1 from treated macrophages were determined by using commercial ELISA kits (R&D Systems, Minneapolis, MN, USA). Briefly, 100 μL of cell-culture medium samples and IL-1β, IL-10, and TGF-β1 protein standards were loaded per well of 96-well plates for further incubation with biotinylated antibodies. Streptavidin-conjugated horseradish peroxidase was next added to each well and followed by substrate solution. Finally, enzyme reactions were stopped and OD_450_ values were measured by Infinite M200 microplate reader (TECAN, Männedorf, Switzerland).

### 2.8. Chromatin Immunoprecipitation (ChIP)

Involvement of NF-κB, IRF3, and STAT-3 in M2-associated gene expressions was validated by ChIP assays. The protocol used was based on the manufacturer’s instruction of EZ-ChIP kit (EMD Millipore). Briefly, macrophages were treated with 1% formaldehyde before cell lysis and DNA fragmentation. After being precleared with protein G-conjugated agarose, a 10 μL aliquot of cell lysate was saved as “input” fraction and the remaining lysate was incubated with control IgG or the antibody against NF-κB (1:100, cat. #1546-1, Epitomics), IRF3 (1:100, cat. #sc-33641, Santa Cruz Biotechnology), or STAT-3 (1:200, cat. #12640, Cell Signaling) for immunoprecipitation. DNA was extracted from the immunoprecipitate for further PCR amplification. The primers and PCR conditions are listed in Appendix A.

### 2.9. shRNA-Mediated Sec22b Knockdown

The empty vector pLKO.1 *puro* and three recombinant plasmids expressing three different 21-mer shRNA sequences against Sec22b mRNA were obtained from the National RNAi Core Facility (Taipei, Taiwan). These plasmids were introduced into THP-1 cells by pseudotyped lentivirus. The infected THP-1 cells were selected against 2 μg/mL of puromycin for 14 days, and cell clones were screened for Sec22b downregulation by RT-PCR and immunoblot analysis. Among the three Sec22b shRNA-expressing plasmids, two (designated as shSec22b #1 and #2) of them efficiently downregulated Sec22b expression by targeting the Sec22b sequences ACTTGCAGCAGTAGCTGTATT and TAGAACCCAGTAGGTGTATAT, respectively. Therefore, the THP-1 cell clones with shSec22b #1 and #2 were chosen for further experiments.

### 2.10. Mouse Tumor Model

The mouse experiment was performed with C57BL/6 mice (~8 weeks old) with permission of the Institutional Animal Care and Use Committee of National Health Research Institutes (NHRI-IACUC-106031-A, 109022-M2-S02, and 109196-A). To evaluate the tumor-promoting capability of eHSP90α-stimulated macrophages, we assayed the tumor-growing rates of Panc 02 cells alone and Panc 02 cells mixed with PBS, LPS, or rHSP90α-treated RAW264.7 cells in C57BL/6 mice, respectively. RAW264.7 cells were incubated for 16 h with 0.5% serum-containing medium and then treated with PBS, 100 ng/mL of LPS, or 15 μg/mL of rHSP90α for another 24 h. For transplantation per mouse, the treated RAW264.7 cells (2.5 × 10^5^) were premixed with 1 × 10^6^ Panc 02 cells in 4-fold-diluted Matrigel (BD Biosciences) before subcutaneous inoculation. The sizes of developing tumors were superficially measured with a Vernier caliper and the estimated tumor volumes were calculated with the formula 1/2 × length × width^2^. Mice were sacrificed on day 30 post-inoculation and tumors were surgically removed for weighing.

### 2.11. Immunohistochemistry (IHC)

The paraffin-embedded tissue sections were deparaffinized by xylene and rehydrated in gradual ethanol dilutions. The sections were subsequently heated for 15 min in 10 mM of citrate buffer, pH 6.0, using a pressure cooker for antigen retrieval. After endogenous peroxidase activity was depleted by 0.3% hydrogen peroxide, the sections were blocked for 30 min with PBS plus 3% bovine serum albumin (BSA) at room temperature in a humidified chamber. The sections were further incubated overnight at 4 °C with the primary antibody against F4/80 (1:100, cat. #MCA497R, AbD Serotec), CD163 (1:80, cat. #sc-33560, Santa Cruz Biotechnology), CD4 (1:100, cat. #GTX44531, GeneTex Inc., Hsinchu City, Taiwan), or CD31 (1:150, cat. #ab28364, Abcam) in PBS plus 0.05% Tween-20 (PBST). After PBST washing, the sections were incubated for 30 min with the corresponding secondary antibodies at room temperature and then dye-stained using the REAL EnVision Detection System (DAKO, Produktionsvej 42, DK-2600 Glostrup, Denmark). After counterstaining with hematoxylin, the stained sections were dehydrated in graded ethanol solutions and xylene, mounted with mounting solution, and finally observed and photographed using the Axiovert S100/AxioCam HR microscope system (Carl Zeiss, Oberkochen, Germany).

### 2.12. Statistical Analysis

The results of cell culture experiments and the mouse model were statistically analyzed by independent samples *t* test. The differences were considered significant if *p* < 0.05.

## 3. Results

### 3.1. eHSP90α Induces M2-Type Polarization of Macrophages

We previously reported that the secretion of macrophages stimulated human pancreatic ductal epithelial HPDE cells to express and secrete HSP90α. To investigate whether macrophage-affected HPDE cells would have a feedback to macrophages, HPDE cells were pretreated with macrophage-conditioned medium (Mϕ-CM) and then their secreted proteins in the medium, designated as HPDE(Mϕ-CM)-CM, were collected to treat macrophages. As shown in Figure 1A, mRNA levels of M1-associated TNF-α and IL-1β were significantly downregulated whereas those of M2-associated CD163, CD204, and IL-10 were significantly upregulated in the THP-1-derived macrophages treated with HPDE(Mϕ-CM)-CM. This phenomenon was antagonized by the presence of anti-HSP90α antibody, suggesting that HSP90α secreted by macrophage-stimulated HPDE cells could act back onto macrophages and induce macrophage M2-polarization. In our previous study, the secretion of EndoMT-derived cells could induce the M2-polarization of THP-1-derived macrophages. We herein showed that EndoMT cell-conditioned medium (EndoMT-CM) was also able to induce M2-associated CD163, CD204, and TGF-β and downregulate M1-associated TNF-α and IL-1β in BMDMs (Figure 1B). The presence of anti-HSP90α antibody antagonized the effects of EndoMT-CM (Figure 1C), suggesting again the ability of eHSP90α to induce macrophage M2 polarization. We therefore investigated the effect(s) of rHSP90α on macrophages, and the results showed that rHSP90α inhibited M1-marker expression but induced M2-associated genes in the macrophages both from human and mouse sources (Figure 1D–G). Furthermore, we investigated the effect of eHSP90α on the phagocytotic activity of macrophages. The phagocytotic activity was evaluated by measuring the number of fluorescent beads engulfed by macrophages. The data showed that the phagocytosis of macrophage was obviously repressed by EndoMT-CM treatment (Figure 1H). Given that eHSP90α is a crucial component of EndoMT-CM, the inhibitory effect of EndoMT-CM on the macrophage phagocytosis was effectively suppressed by the eHSP90α inhibitor DMAG-N-oxide or anti-HSP90α antibody (Figure 1H). Repression of phagocytosis was also observed when macrophages were treated with rHSP90α (Figure 1I). These data together suggest that eHSP90α induces macrophage M2 polarization with a reduction of phagocytotic activity.

### 3.2. eHSP90α Induces Macrophage M2 Markers via CD91 and TLR4 Receptor-Associated MyD88-IRAK1/4−IKKα/β−NF-κB/IRF3 Pathway

We previously reported that TLR4 was associated with CD91 as a cell-surface receptor complex for eHSP90α binding to induce macrophage M2 polarization. As shown in Figure 2A, the TLR4-CD91 complex significantly recruited MyD88 and IRAK1 upon rHSP90α treatment. Cellular phosphorylation (activation) levels of IRAK1 and IRAK4 were also increased, while the increases could be drastically abolished by CD91 or TLR4-antagonizing antibody (Figure 2B), suggesting that a CD91-TLR4-MyD88 complex-associated IRAK−IKK−NF-κB/IRF canonical pathway could be involved in the action mechanism of eHSP90α. Indeed, rHSP90α significantly induced phosphorylation (activation) levels of IKKα/β and the downstream transcriptional factors NF-κB and IRF3 (Figure 2B). An inhibitor of IKKα/β (IKKi) exhibited a repressive effect on rHSP90α-induced phosphorylation (activation) of IKKα/β, IRF3, and NF-κB (Figure 2C). Decreased IL-1β and increased IL-10 and TGF-β secretions by macrophages after rHSP90α treatment were also significantly repressed by the presence of IKKi (Figure 2D). Consistently, IKKi also inhibited the effects of rHSP90α treatment on TNF-α, IL-1β, CD163, CD204, and IL-10 mRNA expressions (Figure 2E). Because putative NF-κB and IRF3-binding sites were recognized on the promoter regions of *CD163*, *CD204*, and *IL-10* genes, we performed chromatin immunoprecipitation (ChIP) assay to confirm whether NF-κB and IRF3 functioned as transcription factors of *CD163*, *CD204*, and *IL-10* genes in macrophages. As shown in Figure 2F,G, rHSP90α treatment induced binding of NF-κB and IRF3 to *CD163*, *CD204*, and *IL-10* gene promoters. Based on these results, we conclude that eHSP90α can induce macrophage M2-marker expression via the MyD88-IRAK1/4−IKKα/β−NF-κB/IRF3 signaling pathway.

### 3.3. CD91 and TLR4 Receptor-Associated MyD88-IRAK1/4−JAK2/TYK2−STAT-3 Pathway Is Also Involved in eHSP90α-Induced Macrophage M2-Polarization

Besides IRAK1, JAK2 and TYK2 were physically associated with MyD88 upon rHSP90α stimulation (Figure 3A). Phosphorylation (activation) levels of JAK2, TYK2, and the downstream transcriptional factor STAT-3 were also induced by rHSP90α, which could be effectively diminished by CD91 or TLR4-antagonizing antibody (Figure 3B), suggesting that a CD91-TLR4−MyD88-IRAK1/4−JAK2/TYK2−STAT-3 pathway was elicited in rHSP90α-treated macrophages. Given that a cytokine receptor-associated JAK2/TYK2−STAT-3 signaling axis is involved in IL-4 and IL-13-induced macrophage M2-polarization, we investigated whether this signaling axis was also responsible for the effects of eHSP90α on macrophages. Our results showed that TNF-α, IL-1β, CD163, CD204, IL-10, and TGF-β mRNA expressions changed by rHSP90α in macrophages could be abrogated by the JAK2 inhibitor JAKi or/and another JAK2/TYK2 inhibitor JSI-124 (Figure 3C). Such observations in THP-1-derived macrophages were consistent with those in the M2-polarization of BMDMs induced by EndoMT-CM or rHSP90α. eHSP90α induced the phosphorylation and activation of JAK2, TYK2, and STAT-3 in BMDMs (Figure 3D,E), whereas JAKi and JSI-124 could effectively block both downregulation of TNF-α and IL-1β mRNA levels and upregulation of CD163, CD204, and IL-10 mRNA expressions induced by EndoMT-CM or rHSP90α (Figure 3F,G). Taken these data together, we suggest that a CD91-TLR4−MyD88-IRAK1/4−JAK2/TYK2–STAT-3 pathway is also involved in eHSP90α-induced macrophage M2 polarization.

### 3.4. CD91 and TLR4 Receptor-Associated JAK2/TYK2−STAT-3−Sec22b Pathway Is Involved in eHSP90α-Inhibited Macrophage Phagocytosis

Next, we investigated whether the CD91 and TLR4 receptor-associated JAK2/TYK2−STAT-3 pathway was also involved in the downregulation of macrophage phagocytosis by eHSP90α. Our data revealed that the inhibitory effect of eHSP90α on macrophage phagocytosis was drastically suppressed by CD91 and TLR4-antagonizing antibodies (Figure 4A) as well as JAK2/TYK2−STAT-3 pathway inhibitors (Figure 4B). Sec22b is a negative regulator of phagocytosis by forming a nonfusogenic complex with Syntaxin 18. A putative STAT-3-binding site was recognized from the promoter region of *sec22b* gene, suggesting *sec22b* as a downstream effector gene of JAK2/TYK2−STAT-3 pathway for the repression of macrophage phagocytosis by eHSP90α. Indeed, Sec22b mRNA expression was induced in rHSP90α-treated macrophages (Figure 4C). In macrophage clones with Sec22b knockdown, the phagocytotic activities were no longer inhibited by rHSP90α (Figure 4D), confirming the involvement of Sec22b in eHSP90α-repressed macrophage phagocytosis. The inhibitors of JAK2/TYK2−STAT-3 pathway repressed rHSP90α-induced Sec22b mRNA expression (Figure 4E), and therefore we performed the ChIP assay to confirm that STAT-3 induced by rHSP90α bound to *sec22b* gene promoter (Figure 4F). These data together suggest that Sec22b is a downstream effector of CD91-TLR4−MyD88-IRAK1/4−JAK2/TYK2−STAT-3 signaling axis and is involved in rHSP90α-repressed macrophage phagocytosis.

### 3.5. eHSP90α-Induced M2-Type Macrophages Exhibit Tumor-Promoting Activity

Given that M2-polarized macrophages exert immunosuppressive and proangiogenic activities to promote tumor growth and metastasis, we first investigated whether eHSP90α-induced M2-type macrophages exhibited a tumor-promoting effect in a mouse model. rHSP90α-treated RAW264.7 cells were mixed with mouse pancreatic adenocarcinoma Panc 02 cells for subcutaneous inoculation into C57BL/6 mice. For comparison with the effect of M1-macrophages, another group of mice were inoculated with Panc 02 cells plus LPS-treated RAW264.7 cells. Growth of the “Panc 02 + rHSP90α-treated RAW264.7” grafts started after day 6 post-inoculation and was faster than those of other groups of grafts (Figure 5A). On day 30 after inoculation, all mice were sacrificed and their tumors were taken for weighing. As shown in Figure 5B, rHSP90α-treated RAW264.7 cells significantly promoted Panc 02 tumor growth, which was in contrast to the tumor-suppressive effect resulted from LPS-treated RAW264.7 cells. Although the tumor growth of “Panc 02” or “Panc 02 + LPS-treated RAW264.7” grafts was much slower than that of “Panc 02 + rHSP90α-treated RAW264.7” grafts, it was noted that abundant Ki-67-positive cells were observed from the small tumor tissues of “Panc 02” and “Panc 02 + LPS-treated RAW264.7” groups as well (Appendix A). The results of IHC analyses revealed that the tumor tissues derived from the “Panc 02 + rHSP90α-treated RAW264.7” grafts contained more CD163^+^ cells but significantly reduced levels of CD4^+^ T-cells when compared with the tumor masses taken from other groups of mice (Figure 5C). We have an undergoing therapeutic study including a cohort of mice receiving the treatments using BMDMs instead of RAW264.7 cells. A consistent result has been obtained, confirming the tumor-promoting characteristic of eHSP90α-treated macrophages. Therefore, our results conclude that eHSP90α-induced M2-polarized macrophages exhibit potent immunosuppressive and tumor-promoting activities.

### 3.6. CD91 and TLR4 Receptor-Associated JAK2/TYK2−STAT-3 Axis Is also Involved in eHSP90α-Induced Proangiogenic Activity of Macrophages

We have already shown a repressed level of CD4^+^ T cells in the tumors promoted by rHSP90α-treated RAW264.7 macrophages. Next, our IHC results revealed that an angiogenic tissue microenvironment was observed in the tumors derived from the Panc 02 plus rHSP90α-treated RAW264.7 cell grafts, which exhibited a higher microvascular density when compared with the tumors taken from other groups of mice (Figure 6A). The study on EndoMT-CM or rHSP90α-stimulated BMDMs showed that VEGF and basic fibroblast growth factor (bFGF) mRNA levels were significantly induced by eHSP90α (Figure 6B,C). In THP-1-derived macrophages, only VEGF mRNA level was induced by rHSP90α through CD91 and TLR4 receptors (Figure 6D). Because the downstream effectors NF-κB and STAT-3 are two well-reported transcriptional regulators of *VEGF* gene expression [29,30], the inhibitor of IKKα/β−NF-κB axis and the inhibitors of JAK2/TYK2−STAT-3 signaling were all as expected to abrogate rHSP90α-induced *VEGF* gene expression in THP-1-derived macrophages (Figure 6E). In EndoMT-CM or rHSP90α-treated BMDMs, eHSP90α-induced VEGF and bFGF mRNA expressions were also effectively abolished by the JAK2/TYK2−STAT-3 pathway inhibitors (Figure 6F,G). These results together conclude that the eHSP90α-induced JAK2/TYK2−STAT-3 pathway is also involved in the elevation of proangiogenic activity in macrophages.

## 4. Discussion

To extend our previous study of eHSP90α-induced macrophage M2-polarization, we first confirmed whether these M2-polarized macrophages exhibited a tumor-promoting capability. We mixed rHSP90α-treated RAW264.7 cells with mouse pancreatic adenocarcinoma Panc 02 cells for subcutaneous inoculation into C57BL/6 mice. In comparison with Panc 02 cells alone or Panc 02 cells mixed with PBS or LPS-treated RAW264.7 cells, only rHSP90α-treated RAW264.7 cells were able to significantly promote the tumor growth of Panc 02 cell grafts. In IHC staining analyses of the tumor tissue sections, we observed a higher microvascular density but a repressed level of CD4^+^ T cells from the tumors derived from Panc 02 plus rHSP90α-treated RAW264.7 cell grafts. This result was consistent with the well-known characteristics of the tumor tissue microenvironment that was infiltrated with abundant M2-polarized macrophages. Therefore, we can conclude that eHSP90α-stimulated macrophages are tumor-promoting M2-macrophages and the tumor-promoting effects can be at least partly attributed by their immunosuppressive and proangiogenic activities.

Furthermore, we studied the signaling transduction pathways elicited in eHSP90α-treated macrophages. Our previous study has indicated that eHSP90α binds with cell-surface CD91 and TLR4 and enhances their physical association and recruitment of MyD88 in macrophages [14]. JAK2 and TYK2 are then recruited onto MyD88 to turn on a MyD88−JAK2/TYK2−STAT-3 signaling pathway to facilitate more HSP90α expression and secretion [14]. In the present study, our data revealed that the JAK2/TYK2−STAT-3 pathway also participates in other events of macrophage M2-polarization, such as downregulation of inflammatory cytokines (TNF-α and IL-1β) as well as upregulation of M2 markers (CD163, CD204, IL-10, and TGF-β), phagocytosis repressor (Sec22b), and angiogenesis activator (VEGF). The JAKs−STATs axes are generally involved in the effects of a wide variety of cytokines [31]. Intrinsically, JAKs are constitutively associated with the intracellular domains of cytokine receptors. The ligations of cytokines with their receptors induce conformational changes of receptors which result in the juxtaposition and phosphorylation of JAKs. The activated JAKs then phosphorylate the cytokine receptors, causing them to recruit STATs for phosphorylation by JAKs. The phosphorylated STATs finally translocate into the nucleus to regulate the downstream gene expression. Besides this canonical format, accumulating evidence has shown that JAKs−STATs axes can also be the downstream signaling events of TLR4-MyD88 complex [25,32,33]. In those macrophages infected with Group A *Streptococcus*, MyD88 is associated with TLR4 and is responsible for STAT-1 phosphorylation and activation by forming a complex with JAK1 and STAT-1 [33]. In LPS-treated macrophages, JAK2 associates with the TLR4-MyD88 complex and participates directly in LPS-induced STAT-5-mediated IL-6 production [25]. Despite the importance of TLR4-MyD88 complex-associated JAK1−STAT-1 and JAK2−STAT-5 pathways in macrophage M1-polarization, our study has highlighted the influential involvement of the TLR4-MyD88 complex-associated JAK2/TYK2−STAT-3 pathway in the M2-polarization and eHSP90α secretion of macrophages. The diverse combinations of four JAK members (JAK1, JAK2, JAK3, and TYK2) and seven STATs (STAT-1, STAT-2, STAT-3, STAT-4, STAT-5a, STAT-5b, and STAT-6) have conferred on JAKs−STATs signal axes a plethora of biological functions [31]. However, the complexity and importance of the JAKs−STATs signaling are increasing as more related upstream stimuli and receptor complexes are disclosed.

In addition to the MyD88−JAK2/TYK2−STAT-3 signaling pathway, eHSP90α also induced a MyD88−IRAK1/4−IKKα/β−NF-κB/IRF3 signaling pathway for macrophage M2-polarization. MyD88-associated IRAKs and toll/IL-1 receptor (TIR) domain-containing adaptor protein (TIRAP) and non-MyD88-associated TIR-domain-containing adaptor inducing IFN-β (TRIF) and TRIF-related adaptor molecule (TRAM) are all associated with TLR4 to trigger signaling cascades resulting in the activation of IKKs and their downstream transcriptional factors NF-κB and IRF3 [25,26]. NF-κB and IRF3 have been known as key transcription factors for macrophage M1-polarization and are responsible for induction of many inflammatory proteins, including TNF-α, IL-1β, IL-6, IL-12p40, IFN-β, and cyclooxygenase-2 [26,27,34]. However, our present study has clearly indicated that an IKK inhibitor repressed eHSP90α-induced NF-κB and IRF3 phosphorylation (activation) as well as the further CD163, CD204, and IL-10 expressions and that eHSP90α facilitated the binding of NF-κB and IRF3 with the promoter regions of CD163, CD204, and IL-10, suggesting that NF-κB and IRF3 are also directly involved in macrophage M2-marker gene expressions. During this process, CD91 and MyD88-associated IRAK1/4−IKKα/β−NF-κB/IRF3 signaling pathway is also required for eHSP90α-induced downregulation of M1 genes. The switch between activation and repression of the inflammatory cytokine expressions remains an enigma. It also needs to be investigated whether the MyD88−JAK2/TYK2−STAT-3 signaling axis has some role(s) herein through a crosstalk with the MyD88−IRAK1/4−IKKα/β−NF-κB/IRF3 signaling pathway. Additionally, the participation of CD91 can be a crucial factor in determining macrophage polarization, given that TLR4 is also the receptor for LPS to induce macrophage M1-type activation [35,36], while CD91 is a negative modulator in this issue and thus CD91 deletion facilitates LPS-treated macrophages to express more M1 events such as higher levels of TNF-α, IL-1β, and IL-6 [37,38]. In eHSP90α-stimulated macrophages, CD91 is enhanced to be physically associated with TLR4 upon eHSP90α binding and probably has more effects on TLR4 and the associated signaling axes.

Our present study has demonstrated that eHSP90α-induced M2-polarized macrophages exhibited less phagocytotic activity. Membrane fusion between the endoplasmic reticulum and plasma membrane is an important membrane supply to facilitate phagocytosis, which is mediated by soluble N-ethylmaleimide-sensitive factor attachment protein (SNAP) receptors (SNAREs) extended from each membrane [39]. Syntaxin 18 is a member of the SNARE family involved in the membrane fusion during macrophage phagocytosis, but Sec22b functions as an inhibitory SNARE to form a complex with Syntaxin 18 and thus inhibits Syntaxin-18-mediated membrane fusion and macrophage phagocytosis [40]. Our data suggest that eHSP90α-induced STAT-3 activation can activate *sec22b* gene expression to further repress the phagocytotic activity of macrophages. Phagocytosis is the process by which macrophages capture and engulf microbes or small particles. Interestingly, efferocytosis is a phagocytosis-like but distinct process for clearance of apoptotic cells and is mediated by different receptors, bridging molecules, and downstream signaling pathways [41]. Cell-surface neuropilin-2 has been reported to promote efferocytosis and facilitate tumor growth [42]. We have preliminarily observed that eHSP90α-stimulated macrophages express elevated levels of neuropilin-1 and neuropilin-2. The effects of eHSP90α on macrophage efferocytosis need further investigation.

## 5. Conclusions

In summary, our present study has demonstrated a tumor-promoting activity exhibited by eHSP90α-induced M2-polarized macrophages. In these macrophages, CD91 and TLR4 receptor complex-associated MyD88−IRAK1/4−IKKα/β−NF-κB/IRF3 and MyD88−JAK2/TYK2−STAT-3 signaling pathways are elicited to cause several events for tumor-promoting M2-polarization, such as downregulation of inflammatory cytokines as well as upregulation of M2 markers, phagocytosis repressors, and angiogenesis activators (Figure 7). Several aspects such as excess eHSP90α, ligation of eHSP90α with CD91-TLR4 complex, and the pathways driving the M2 polarization can serve as targets for developing novel cancer preventive and therapeutic strategies.

## Figures and Tables

**Figure 1 cells-11-00229-f001:**
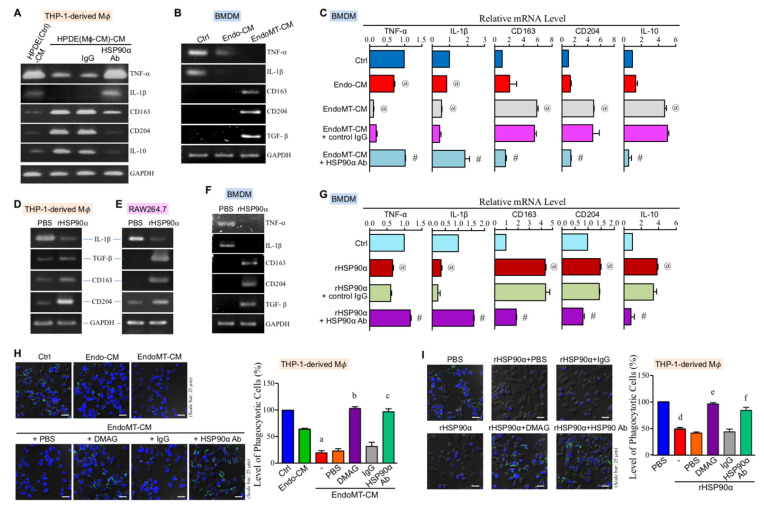
eHSP90α induces M2-type polarization of macrophages. (**A**) mRNA levels of TNF-α, IL-1β, CD163, CD204, and IL-10 in the THP-1-derived macrophages treated for 24 h with HPDE(Ctrl)-CM or HPDE(Mϕ-CM)-CM in the absence or presence of 10 μg/mL control IgG or anti-HSP90α antibody (LTK Biotechnologies, Taoyuan, Taiwan [14]). The representative RT-PCR data of three independent experiments are shown. (**B**) mRNA levels of TNF-α, IL-1β, CD163, CD204, and TGF-β in the BMDMs treated for 24 h with control medium (Ctrl), Endo-CM, or EndoMT-CM. The representative RT-PCR data of three independent experiments are shown. (**C**) mRNA levels of TNF-α, IL-1β, CD163, CD204, and IL-10 in the BMDMs treated for 24 h with control medium (Ctrl), Endo-CM, EndoMT-CM, or EndoMT-CM plus 10 μg/mL of control IgG or anti-HSP90α antibody. The mean ± SD of qPCR data were obtained from three independent experiments. ^@^
*p* < 0.01 when compared with “Ctrl” group. ^#^
*p* < 0.01 when compared with “EndoMT-CM + control IgG” group. (**D**,**E**) mRNA levels of IL-1β, TGF-β, CD163, and CD204 in the THP-1-derived macrophages (**D**) or RAW264.7 cells (**E**) treated for 24 h with PBS or 15 μg/mL of rHSP90α. The representative RT-PCR data of three independent experiments are shown. (**F**) mRNA levels of TNF-α, IL-1β, CD163, CD204, and TGF-β in the BMDMs treated for 24 h with PBS or 15 μg/mL of rHSP90α. The representative RT-PCR data of three independent experiments are shown. (**G**) mRNA levels of TNF-α, IL-1β, CD163, CD204, and IL-10 in the BMDMs treated for 24 h with PBS (Ctrl), 15 μg/mL of rHSP90α, or rHSP90α plus control IgG or anti-HSP90α antibody. The qPCR data shown are the mean ± SD of three independent experiments. ^@^
*p* < 0.01 when compared with “Ctrl” group. ^#^
*p* < 0.01 when compared with “rHSP90α + control IgG” group. (**H**) Phagocytotic activities of the THP-1-derived macrophages treated for 24 h with control medium (Ctrl), Endo-CM, EndoMT-CM, or EndoMT-CM plus PBS, 1 μM of DMAG-N-oxide, control IgG, or anti-HSP90α antibody. The quantitative data are the mean ± SD of three independent experiments. ^a^
*p* < 0.01 when compared with “Ctrl” group. ^b^
*p* < 0.01 when compared with “EndoMT-CM + PBS” group. ^c^
*p* < 0.01 when compared with “EndoMT-CM + IgG” group. (**I**) Phagocytotic activities of the THP-1-derived macrophages treated for 24 h with PBS or rHSP90α in the absence or presence of DMAG-N-oxide or anti-HSP90α antibody. The quantitative data are the mean ± SD of three independent experiments. ^d^
*p* < 0.01 when compared with “PBS” group. ^e^
*p* < 0.01 when compared with “rHSP90α + PBS” group. ^f^
*p* < 0.01 when compared with “rHSP90α + IgG” group.

**Figure 2 cells-11-00229-f002:**
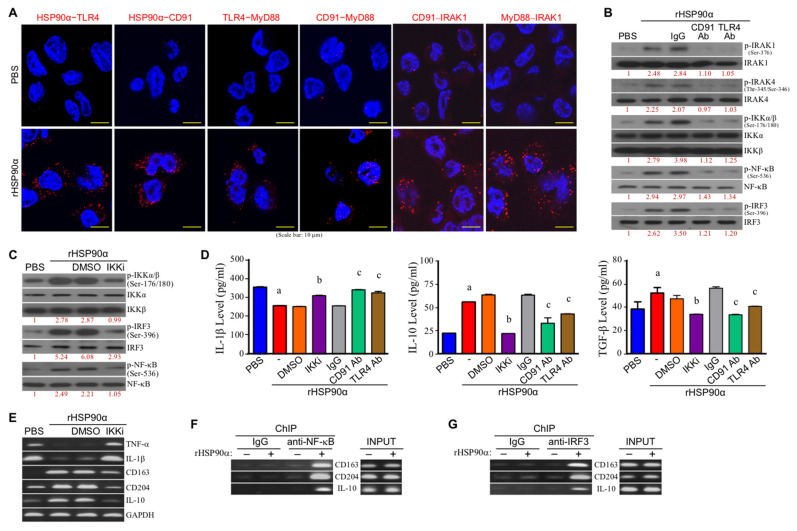
The canonical CD91/MyD88–IRAK1/4–IKKα/β–NF-κB/IRF3 signaling pathway is involved in eHSP90α-induced macrophage M2-polarization. (**A**) Red fluorescent dots in PLAs showed that the physical associations of HSP90α–TLR4, HSP90α–CD91, TLR4–MyD88, CD91–MyD88, CD91–IRAK1, and MyD88–IRAK1 were induced when THP-1-derived macrophages were treated for 4 h with 15 μg/mL of rHSP90α. The representative data of three independent experiments are shown. (**B**) Levels of IRAK1, phosphorylated IRAK1, IRAK4, phosphorylated IRAK4, IKKα, IKKβ, phosphorylated IKKα/β, NF-κB, phosphorylated NF-κB, IRF3, and phosphorylated IRF3 in the THP-1-derived macrophages treated with PBS, rHSP90α, or rHSP90α plus control IgG or anti-CD91 or TLR4 antibody. The immunoblot data shown are the representative of three independent experiments. The numbers below the image sets are the relative levels of the phosphorylated proteins after normalization to their respective protein total levels. (**C**) Levels of IKKα, IKKβ, phosphorylated IKKα/β, IRF3, phosphorylated IRF3, NF-κB and phosphorylated NF-κB in the THP-1-derived macrophages treated with PBS, rHSP90α, or rHSP90α plus DMSO or 0.1 μM of IKKα/β inhibitor (IKKi; 6-amino-4-(4-phenoxyphenylethylamino)quinazoline, cat. #481406, Merck KGaA, Darmstadt, Germany [28]). The immunoblot data shown are the representative of three independent experiments. Relative phosphorylated protein levels are indicated below the image sets. (**D**) Secreted levels of IL-1β, IL-10, and TGF-β in the THP-1-derived macrophages treated with PBS, rHSP90α, or rHSP90α plus DMSO, IKKi, control IgG, or anti-CD91 or TLR4 antibody. The mean ± SD of ELISA data were obtained from three independent experiments. ^a^
*p* < 0.01 when compared with “PBS” group. ^b^
*p* < 0.01 when compared with “rHSP90α + DMSO” group. ^c^
*p* < 0.01 when compared with “rHSP90α + IgG” group. (**E**) mRNA levels of TNF-α, IL-1β, CD163, CD204, and IL-10 in the THP-1-derived macrophages treated with PBS, rHSP90α, or rHSP90α plus DMSO or IKKi. The RT-PCR data shown are the representative of three independent experiments. (**F**,**G**) ChIP assays showed that rHSP90α induced binding of NF-κB (**F**) and IRF3 (**G**) to *CD163*, *CD204*, and *IL-10* gene promoters. The representative data of three independent experiments are shown.

**Figure 3 cells-11-00229-f003:**
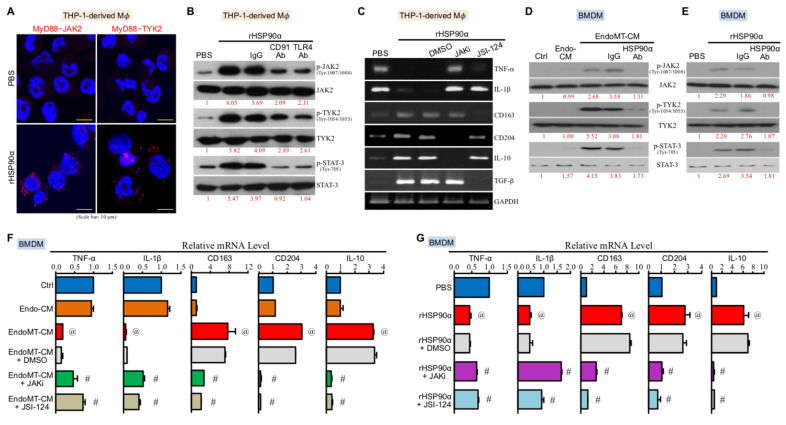
A CD91 and TLR4 receptor-associated MyD88–JAK2/TYK2–STAT-3 signaling axis is involved in eHSP90α-induced macrophage M2-polarization. (**A**) Red fluorescent dots in PLAs showed that the physical associations of MyD88–JAK2 and MyD88–TYK2 were induced when THP-1-derived macrophages were treated for 4 h with 15 μg/mL of rHSP90α. The representative data of three independent experiments are shown. (**B**) Levels of JAK2, phosphorylated JAK2, TYK2, phosphorylated TYK2, STAT-3, and phosphorylated STAT-3 in the THP-1-derived macrophages treated for 4 h with PBS, 15 μg/mL of rHSP90α, or rHSP90α plus control IgG or anti-CD91 or TLR4 antibody. The immunoblot data shown are the representative of three independent experiments. Relative levels of the phosphorylated proteins are indicated below the image sets after normalization to the respective protein total levels. (**C**) mRNA levels of TNF-α, IL-1β, CD163, CD204, IL-10, and TGF-β in the THP-1-derived macrophages treated for 24 h with PBS, rHSP90α, or rHSP90α plus DMSO, 10 nM JAK2 inhibitor (JAKi; cat. #420099, Merck KGaA), or 10 μM JSI-124 (Cucurbitacin I, another JAK2/TYK2 inhibitor; cat. #238590, Merck KGaA). The representative RT-PCR data of three independent experiments are shown. (**D**) Levels of JAK2, phosphorylated JAK2, TYK2, phosphorylated TYK2, STAT-3, and phosphorylated STAT-3 in the BMDMs treated 4 h with control medium (Ctrl), Endo-CM, EndoMT-CM, or EndoMT-CM plus control IgG or anti-HSP90α antibody. The immunoblot data shown are the representative of three independent experiments. Relative levels of the phosphorylated proteins were denoted below the image sets after normalization to the respective protein total levels. (**E**) Levels of JAK2, phosphorylated JAK2, TYK2, phosphorylated TYK2, STAT-3, and phosphorylated STAT-3 in the BMDMs treated 4 h with PBS, rHSP90α, or rHSP90α plus control IgG or anti-HSP90α antibody. The immunoblot data shown are the representative of three independent experiments. Relative phosphorylated protein levels are indicated below the image sets. (**F**) mRNA levels of TNF-α, IL-1β, CD163, CD204, and IL-10 in the BMDMs treated for 24 h with control medium (Ctrl), Endo-CM, EndoMT-CM, or EndoMT-CM plus DMSO, JAKi, or JSI-124. The mean ± SD of qPCR data were obtained from three independent experiments. ^@^
*p* < 0.01 when compared with “Ctrl” group. ^#^
*p* < 0.01 when compared with “EndoMT-CM + DMSO” group. (**G**) mRNA levels of TNF-α, IL-1β, CD163, CD204, and IL-10 in the BMDMs treated for 24 h with PBS, rHSP90α, or rHSP90α plus DMSO, JAKi, or JSI-124. The qPCR data shown are the mean ± SD of three independent experiments. ^@^
*p* < 0.01 when compared with “PBS” group. ^#^
*p* < 0.01 when compared with “rHSP90α + DMSO” group.

**Figure 4 cells-11-00229-f004:**
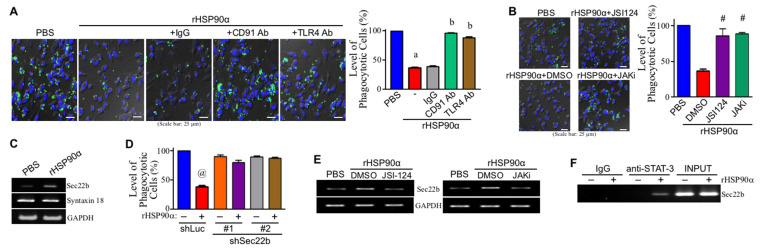
The MyD88–JAK2/TYK2–STAT-3–Sec22b pathway is involved in the repression of macrophage phagocytosis by eHSP90α. (**A**) Phagocytotic activities of the THP-1-derived macrophages treated for 24 h with PBS, rHSP90α, or rHSP90α plus control IgG or anti-CD91 or TLR4 antibody. The quantitative data are the mean ± SD of three independent experiments. ^a^
*p* < 0.01 when compared with “PBS” group. ^b^
*p* < 0.01 when compared with “rHSP90α + IgG” group. (**B**) Phagocytotic activities of the THP-1-derived macrophages treated with PBS or rHSP90α plus DMSO, JSI-124, or JAKi. The quantitative data are the mean ± SD of three independent experiments. ^#^
*p* < 0.01 when compared with “rHSP90α + DMSO” group. (**C**) mRNA levels of Sec22b and Syntaxin 18 in the THP-1-derived macrophages treated for 24 h with PBS or 15 μg/mL of rHSP90α. The representative RT-PCR data of three independent experiments are shown. (**D**) Phagocytotic activities of the Sec22b-knockdown macrophages treated with PBS or rHSP90α. Two THP-1 cell clones expressing Sec22b-targeting shRNA #1 and #2, respectively, were induced by 12-*O*-tetradecanoyl-13-phorbol acetate to differentiate into macrophages. The macrophages were then treated with PBS or rHSP90α for 24 h. The quantitative data are the mean ± SD of three independent experiments. ^@^
*p* < 0.01 when compared with “–” group. (**E**) Sec22b mRNA levels in the THP-1-derived macrophages treated for 24 h with PBS or rHSP90α plus DMSO, JSI-124, or JAKi. The RT-PCR data shown are the representative of three independent experiments. (**F**) ChIP assay showing rHSP90α-induced binding of STAT-3 to *sec22b* gene promoter. The representative data of three independent experiments is shown.

**Figure 5 cells-11-00229-f005:**
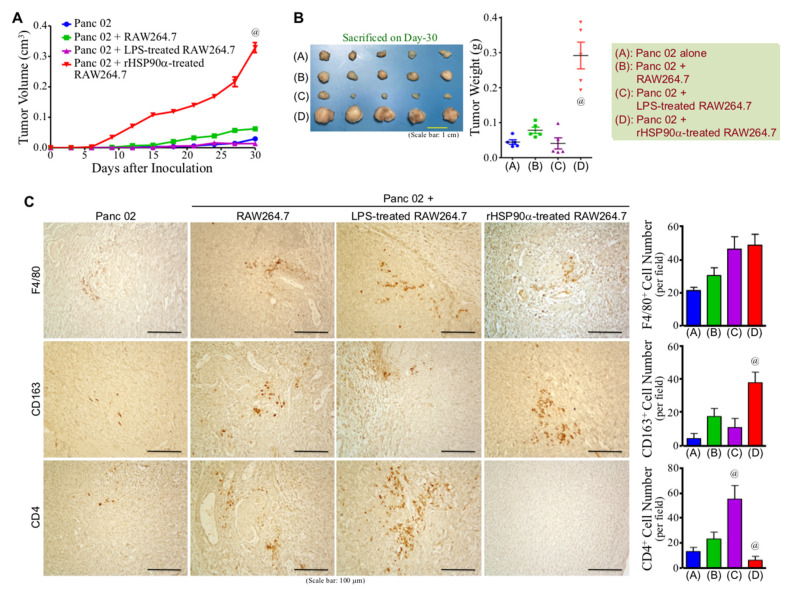
eHSP90α-induced M2-polarized macrophages exhibit significant tumor-promoting activity. (**A**,**B**) Promotion of the tumor growth of Panc 02 cell grafts by rHSP90α-treated macrophages. C57BL/6 mice were subcutaneously injected with Panc 02 cells alone or together with the RAW264.7 cells pretreated for 24 h with 100 ng/mL of LPS or 15 μg/mL of rHSP90α (*n* = 5 per group). The sizes of developing tumors were superficially measured using a Vernier caliper and their volumes were calculated with the formula of ½ × length × width^2^ (**A**). Mice were sacrificed on day 30 post-inoculation and tumors were removed (**B**). ^@^
*p* < 0.01 when compared with “Panc 02” or “Panc 02 + RAW264.7”. (**C**) IHC of F4/80, CD163, and CD4 from the tumors formed by Panc 02 cells alone (designated as “Panc 02” group) or Panc 02 cells mixed with PBS-, LPS-, or rHSP90α-treated RAW264.7 cells (designated as “Panc 02 + RAW264.7”, “Panc 02 + LPS-treated RAW264.7”, and “Panc 02 + rHSP90α-treated RAW264.7” groups, respectively). Quantification of IHC was performed by counting positively stained cells from 3~5 high power fields of each tissue section. ^@^
*p* < 0.05 when compared with “Panc 02” or “Panc 02 + RAW264.7” group.

**Figure 6 cells-11-00229-f006:**
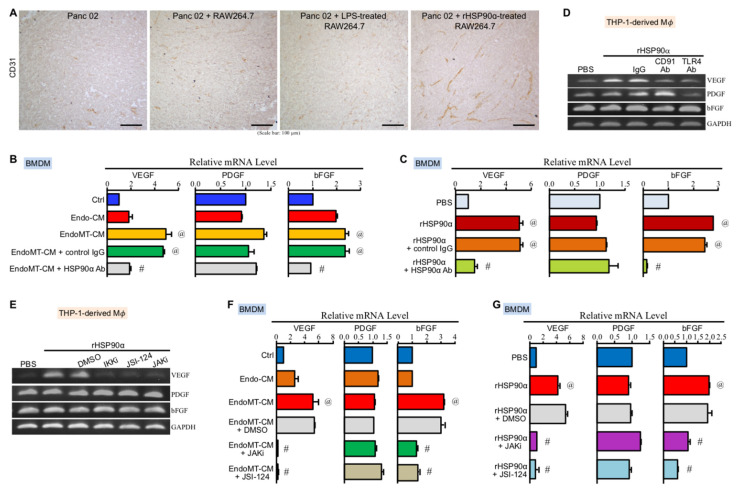
The MyD88–JAK2/TYK2–STAT-3 axis is involved in eHSP90α-induced proangiogenic activity of M2-type macrophages. (**A**) IHC of CD31 from the tumors formed by Panc 02 cells alone or Panc 02 cells mixed with PBS-, LPS-, or rHSP90α-treated BMDMs. (**B**) mRNA levels of VEGF, platelet-derived growth factor (PDGF), and bFGF in the BMDMs treated for 24 h with control medium (Ctrl), Endo-CM, EndoMT-CM, or EndoMT-CM plus control IgG or anti-HSP90α antibody. The mean ± SD of qPCR data were obtained from three independent experiments. ^@^
*p* < 0.01 when compared with “Ctrl” group. ^#^
*p* < 0.01 when compared with “EndoMT-CM + control IgG” group. (**C**) mRNA levels of VEGF, PDGF, and bFGF in the BMDMs treated for 24 h with PBS, rHSP90α, or rHSP90α plus control IgG or anti-HSP90α antibody. The qPCR data shown are the mean ± SD of three independent experiments. ^@^
*p* < 0.01 when compared with “PBS” group. ^#^
*p* < 0.01 when compared with “rHSP90α + control IgG” group. (**D**) mRNA levels of VEGF, PDGF, and bFGF in the THP-1-derived macrophages treated for 24 h with PBS, rHSP90α, or rHSP90α plus control IgG or anti-CD91 or TLR4 antibody. The representative RT-PCR data of three independent experiments are shown. (**E**) mRNA levels of VEGF, PDGF, and bFGF in the THP-1-derived macrophages treated for 24 h with PBS, rHSP90α, or rHSP90α plus DMSO, IKKi, JSI-124, or JAKi. The RT-PCR data shown are the representative of three independent experiments. (**F**) mRNA levels of VEGF, PDGF, and bFGF in the BMDMs treated for 24 h with control medium (Ctrl), Endo-CM, EndoMT-CM, or EndoMT-CM plus DMSO, JAKi, or JSI-124. The qPCR data shown are the mean ± SD of three independent experiments. ^@^
*p* < 0.01 when compared with “Ctrl” group. ^#^
*p* < 0.01 when compared with “EndoMT-CM + DMSO” group. (**G**) mRNA levels of VEGF, PDGF, and bFGF in the BMDMs treated for 24 h with PBS, rHSP90α, or rHSP90α plus DMSO, JAKi, or JSI-124. The qPCR data shown are the mean ± SD of three independent experiments. ^@^
*p* < 0.01 when compared with “PBS” group. ^#^
*p* < 0.01 when compared with “rHSP90α + DMSO” group.

**Figure 7 cells-11-00229-f007:**
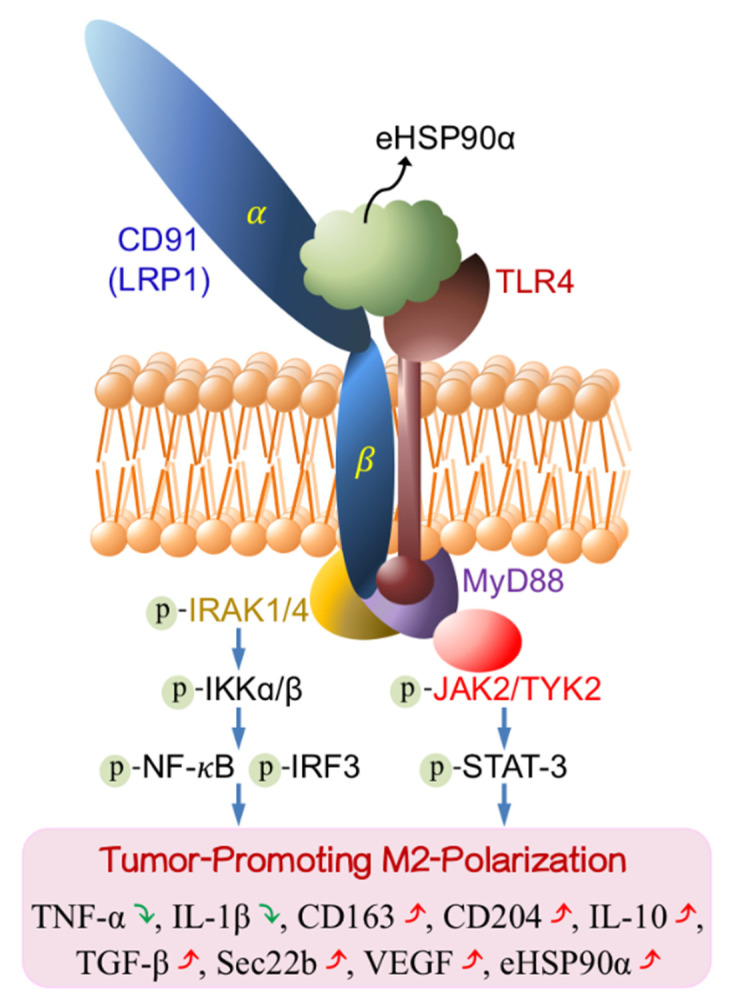
Schematic summary of eHSP90α-induced CD91 and TLR4-associated signal transduction pathways for tumor-promoting M2-polarization in macrophages. Once macrophages are exposed to eHSP90α, CD91 is enhanced to be physically associated with TLR4 to recruit IRAK1/4 and MyD88. Besides the canonical CD91/MyD88–IRAK1/4–IKKα/β–NF-κB/IRF3 signaling pathway, JAK2 and TYK2 are recruited onto MyD88 to turn on a MyD88−JAK2/TYK2−STAT-3 pathway. Both signaling axes act together leading to downregulation of inflammatory cytokines TNF-α and IL-1β as well as upregulation of M2-associated TGF-β, CD163, CD204, IL-10, and VEGF expressions. Following our previous report that the eHSP90α-induced MyD88−JAK2/TYK2−STAT-3 pathway is involved in a feedforward loop of HSP90α expression and secretion, this signal transduction pathway is also responsible for the induction of Sec22b expression to decrease the phagocytotic activity of macrophages. Green ↓, down-regulated. Red ↑, up-regulated.

## Data Availability

Not applicable.

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
