# Peer review of "Extracellular HSP90α Induces MyD88-IRAK Complex-Associated IKKα/β−NF-κB/IRF3 and JAK2/TYK2−STAT-3 Signaling in Macrophages for Tumor-Promoting M2-Polarization"

_cells, 2022, doi:10.3390/cells11020229_

Round 1
Reviewer 1 Report
In this manuscript, Fan et al. investigated macrophage M2-polarization effect induced by Extracellular HSP90a (eHSP90a), demonstrated that eHSP90a induces M2-type polarization of macrophages which exhibit significant tumor-promoting activity. Then they evaluated that both MyD88-IRAK1/4-IKKa/b-NF-kB/IRF3 pathway and MyD88-IRAK1/4−JAK2/TYK2−STAT-3 pathway are involved in eHSP90-induced macrophage M2-polarization. In addition, they found the Sec22b is also involved in rHSP90a-repressed macrophage phagocytosis. This manuscript is very interesting and lays down useful information. In general, the manuscript was well written.
Author Response
We thank the reviewer’s positive recommendation on this submission.
Reviewer 2 Report
The authors described a role for extracellular HSP90 to polarize macrophage toward an M2 phenotype. The argument is of interest and deserves accurate research. However, it is to note that part of the results described in the manuscript lacks novelty (the authors published in 2019 a paper showing similar data, see PMID: 31847880).
In addition, the manuscript lacks a common thread from an experimental point of view and several concerns must be addressed:
1-It is not clear how EndoMT-derived cells have been generated.
2- In vitro experiments with conditioned medium using PANC2 and RAW264.7 are required to introduce the in vivo mouse model.
3-To show eHSP90 level in cancer cell and macrophage conditioned medium, for instance by western blot, will clarify the ability of these cells to release HSP90. This is particularly important for the cell lines used in the in vivo experiment. Indeed, both cancer cells and macrophages can secrete HSP90 and as far as I know RAW cells already secrete high HSP90 levels.
5-PANC2 cells (1*106 injected subcutaneously) seem to grow very poorly in absence of RAW treated cells. To repeat the experiments with a different cancer mouse model is mandatory (using different cancer cells).
6-Please show the results related to the experiment performed with BMDMs instead of RAW264.7 cells (see lines 358-360, page 9).
7-Due to the uneven distribution of immune cells in the tumor, a quantification of positive cells in the IHC is required (Figure 2c). To generate a more comprehensive and exhaustive analysis of immune cell populations and endothelial cells (Figure 6a) inside the tumor, a cytofluorimetric analysis is strongly suggested.
8-Apparently, the authors used the same antibodies against TLR4 (a wrong Santa Cruz code is indicated) and LRP1 (recognizing an extracellular antigen) for proximity assays with both extracellular (rHSP90) and intracellular (IRAK1 and MyD88) interactors. This is unlikely.
9-Please clearly identify the cell types in Figure 3 (are there BMDM or THP1-derived macrophage or RAW264.7?)
10-Please specify the name and reference of the IKK inhibitor used in Figure 3 and JAK2 inhibitor.
11-Please show quantification and statistics for all the real time analysis and western blot present in the manuscript.
Author Response
General Comment: “The authors described a role for extracellular HSP90 to polarize macrophage toward an M2 phenotype. The argument is of interest and deserves accurate research. However, it is to note that part of the results described in the manuscript lacks novelty (the authors published in 2019 a paper showing similar data, see PMID: 31847880). In addition, the manuscript lacks a common thread from an experimental point of view and several concerns must be addressed…”
Response: The work of this submission is a part of our series of studies. In our previous study published in J Hematol Oncol 2019, we reported that endothelial-mesenchymal transition (EndoMT)-derived cells secreted HSP90α to induce macrophage M2-polarization through cell-surface CD91 and TLR4 receptors. The secreted extracellular HSP90α (eHSP90α) did not only induce expressions of several M2 markers, but also induced a large amount of HSP90α expressed and secreted from the M2-polarized macrophages to create an eHSP90α-rich tissue microenvironment for promoting tumor growth. In the further study published in Cells 2020, we showed that octyl gallate exhibited a repressive effect on EndoMT-promoted and M2-macrophage-involved tumor growth by blocking eHSP90α–TLR4 ligation and thus prevented eHSP90α-induced macrophage M2-polarization and more HSP90α secretion from macrophages and cancer cells. In the current manuscript, we presented an in vivo validation of the tumor-promoting activity exhibited by the eHSP90α-induced M2-polarized macrophages, and furthermore, we tried to disclose the CD91 and TLR4 downstream pathways accounting for the tumor-promoting M2-polarization. This is an extending work of our previous publications, and therefore the data of Figure 1B~1G include the repetitive results summarizing the different conditions mentioned in our previous studies.
Comment 1: “It is not clear how EndoMT-derived cells have been generated.”
Response: Please see the statement in lines 171-178 and the reference #14 coded therein.
Comment 2: “In vitro experiments with conditioned medium using PANC2 and RAW264.7 are required to introduce the in vivo mouse model.”
Response: As mentioned already, one of the research aims of this study is to validate whether eHSP90α-induced M2-polarized macrophages exhibited tumor-promoting activity or not. Certainly, a mix of cancer cells and eHSP90α-treated macrophages should be the first choice for in vivo mouse model.
Comment 3: “To show eHSP90 level in cancer cell and macrophage conditioned medium, for instance by western blot, will clarify the ability of these cells to release HSP90. This is particularly important for the cell lines used in the in vivo experiment. Indeed, both cancer cells and macrophages can secrete HSP90 and as far as I know RAW cells already secrete high HSP90 levels.”
Response: We did measure and compare the eHSP90α levels in the conditioned media of treated cancer cells and macrophages by western blot or/and ELISA in our previously published works. The aims of the current study are to validate whether eHSP90α-induced M2-polarized macrophages exhibited tumor-promoting activity and to investigate the signaling pathways in eHSP90α-stimulated macrophages.
Comment 4: (skipped)
Comment 5: “PANC2 cells (1*106 injected subcutaneously) seem to grow very poorly in absence of RAW treated cells. To repeat the experiments with a different cancer mouse model is mandatory (using different cancer cells).
Response: Mouse pancreatic adenocarcinoma cell line is rare in the world but Panc 02 is currently the only one well-recognized and used pancreatic adenocarcinoma cell line derived from C57BL/6 mouse. Despite we had published a paper using LSL-K-RasG12D/Pdx1-Cre transgenic mice to demonstrate that macrophages and the secreted HSP90α are involved in pancreatic adenocarcinoma development (Oncoimmunology 7: e1424612, 2018), we totally agree with the reviewer’s comment and are planning to use cell lines of other cancer types to confirm the result in the future.
Comment 6: “Please show the results related to the experiment performed with BMDMs instead of RAW264.7 cells (see lines 358-360, page 9).”
Response: In this cohort of mice, we do not only use BMDMs instead of RAW264.7 cells, but we are also performing therapeutic evaluations and the related analyses. The whole experiments will be completed and published via another manuscript submission. However, we are glad to provide a part of data for the reviewers’ review only. An attached data file will be sent via the editorial office.
Comment 7: “Due to the uneven distribution of immune cells in the tumor, a quantification of positive cells in the IHC is required (Figure 2c). To generate a more comprehensive and exhaustive analysis of immune cell populations and endothelial cells (Figure 6a) inside the tumor, a cytofluorimetric analysis is strongly suggested.”
Response: A new Figure 2 with the quantification of IHC and the related figure legend are included in the revised manuscript. Additionally, the data of Figure 6A is to show an obvious increase of microvascular density but not free endothelial cells in the tumors promoted by eHSP90α-induced M2-polarized macrophages.
Comment 8: “Apparently, the authors used the same antibodies against TLR4 (a wrong Santa Cruz code is indicated) and LRP1 (recognizing an extracellular antigen) for proximity assays with both extracellular (rHSP90) and intracellular (IRAK1 and MyD88) interactors. This is unlikely.”
Response: We have corrected the typing error of the catalogue number of TLR4 antibody. However, we do not agree with the reviewer’s “This is unlikely” comment. Generally, the thickness of cell-membrane lipid bilayers is 36~40 Å which corresponds to the average length (approx. 28 amino acid residues) of the transmembrane domains of receptor tyrosine kinases. TLRs are characterized by their relatively long transmembrane helices with an average length of 33-residue or 47 Å. This length is far smaller than 400 Å (40-nm), a limited distance for PLA detection.
Comment 9: “Please clearly identify the cell types in Figure 3 (are there BMDM or THP1-derived macrophage or RAW264.7?).”
Response: Please see the Figure 3 legend. The “THP-1-derived macrophages” has been already indicated several times.
Comment 10: “Please specify the name and reference of the IKK inhibitor used in Figure 3 and JAK2 inhibitor.”
Response: The extra information of IKK inhibitor has been added into the Figure 3 legend. A new reference has also been added as reference #42. The additional information for JAK2 inhibitor has been added into the Figure 4 legend.
Comment 11: “Please show quantification and statistics for all the real time analysis and western blot present in the manuscript.”
Response: New Figure 3B, 3C, 4B, 4D, and 4E with the quantification of western blot and the related figure legends have been included in the revised manuscript.
Reviewer 3 Report
In the manuscript titled “Extracellular HSP90a Induces MyD88−IRAK Complex-Associated IKKa/b−NF-kB/IRF3 and JAK2/TYK2−STAT-3 Signaling in Macrophages for Tumor-Promoting M2-Polarization”, Chi-Shuan Fan et al. expanded their previous published results on the relationship among HSP90a, macrophages and tumors. Here they documented that secreted extracellular HSP90a induces an M2 polarization of macrophages that exhibit strong tumor-promoting activity. More importantly they documented that eHSP90a induced M2 polarization via CD91 and TLR4 associated activation of the MyD88-IRAK1/4-IKKa/b-NF-kB/IRF3 pathway and suggested that, following interaction of eHSP90a with CD91/TLR4, NF-kB and IRF3 functioned as transcription factors for CD163, CD204 and IL-10 genes. In addition, via CD91 and TLR4 associated activation of the JAK2/TYK2-STAT-3 pathway, eHSP90a inhibited macrophage phagocytosis and exerted a proangiogenic activity. Experiments are well conducted and results seem robust.
Specific comments to Authors:
Results of Figure 2C should be accompanied by quantification of the stained cells (numbers of F4/80+, CD163+,and CD4+ cells/HPF) and by the corresponding statistical analysis among the groups.
Results of mice receiving BMDCs instead of RAW261.7 should be shown.
Author Response
General Comments: “In the manuscript titled “Extracellular HSP90a Induces MyD88−IRAK Complex-Associated IKKa/b−NF-kB/IRF3 and JAK2/TYK2−STAT-3 Signaling in Macrophages for Tumor-Promoting M2-Polarization”, Chi-Shuan Fan et al. expanded their previous published results on the relationship among HSP90a, macrophages and tumors. Here they documented that secreted extracellular HSP90a induces an M2 polarization of macrophages that exhibit strong tumor-promoting activity. More importantly they documented that eHSP90a induced M2 polarization via CD91 and TLR4 associated activation of the MyD88-IRAK1/4-IKKa/b-NF-kB/IRF3 pathway and suggested that, following interaction of eHSP90a with CD91/TLR4, NF-kB and IRF3 functioned as transcription factors for CD163, CD204 and IL-10 genes. In addition, via CD91 and TLR4 associated activation of the JAK2/TYK2-STAT-3 pathway, eHSP90a inhibited macrophage phagocytosis and exerted a proangiogenic activity. Experiments are well conducted and results seem robust.”
Response: We thank the reviewer’s positive comments on this submission.
Comment 1: “Results of Figure 2C should be accompanied by quantification of the stained cells (numbers of F4/80+, CD163+, and CD4+ cells/HPF) and by the corresponding statistical analysis among the groups.”
Response: A new Figure 2 with the quantification of IHC and the related figure legend have been included in the revised manuscript.
Comment 2: “Results of mice receiving BMDCs instead of RAW261.7 should be shown.”
Response: Same as the response to the reviewer #2. In this cohort of mice, we do not only use BMDMs instead of RAW264.7 cells, but we are also performing therapeutic evaluations and the related analyses. The whole experiments will be completed and published in another manuscript submission. However, we are glad to provide a part of data for the reviewers’ review only. An attached data file will be sent via the editorial office.
Round 2
Reviewer 2 Report
The authors did not answer the requests of this reviewer.
Author Response
We have reorganized the Results section, added some statements, and also added a data as Supplementary Figure S1. The original Figure 2 is changed to Figure 5. The original Figures 3-5 have been changed to Figures 2-4. Additionally, we have corrected the incomplete description of the PLA method and also added some statements in the figure legends accordingly. Specifically,
Comment 1: “The Authors must depict, in all the figure legends, how many times each shown experiment has been repeated.”
Response: We have added the requested statements in the figure legends.
Comment 2: “The Authors need to quantify the WB bands on a set of at least three experiments and must delineate these quantitative and normalized data in separate representative histograms accompanied by Standard deviation bars and statistics.”
Response: We did quantify the immunoblot images obtained from at least three independent experiments. Despite that the results of different experiments showed the same trend, the quantitative data had big SD values and did not reach a level of statistical significance. But in our manuscript, we did use the related kinase inhibitors to treat macrophages and do other experiments like ELISA (Fig. 2D), RT-PCR (Fig. 2E, Fig. 3C, Fig. 3F, and Fig. 3G), and chromatin immunoprecipitation (Fig. 2F and Fig. 2G) together to draw the experimental conclusion.
Comment 3: “As evidenced in the previous comments raised by the Reviewer 2, the Authors need to show in vitro experiments with conditioned medium using at least one of the two indicate cell lines (PANC2 or RAW264.7 cells). Alternatively, the Authors must carefully explain/discuss in the manuscript why these experiments are not necessary for this study.”
Response: Tumor immunosuppression and angiogenesis are two main tumor-promoting mechanisms contributed by M2-macrophages. Therefore, this study focused on the demonstration whether eHSP90a-induced M2-polarized macrophages exhibit tumor-promoting activity involving these two regards. We still do not understand what in vitro experiments the editor or reviewer asked us to point out. But we have reorganized the Results section, made more statements (lines 433-435 and lines 444-448), and also added a Supplementary Figure S1 data to strengthen the rationale of our in vivo experiments. The original Figure 2 is changed as Figure 5. We hope the combination of Figure 5, Figure 6, and the new Supplementary Figure S1 could help our demonstration more clearer.
Comment 4: “The Authors need to explain how they discriminate between intracellular and secreted HSP90 in their Proximity Ligation Assay experiments.”
Response: I apologized for making an incomplete and misleading statement about the PLA method in the last revised manuscript. The original statement “The treated macrophages werethen fixed with 3% paraformaldehyde, permeabilized with 0.1% Triton X-100, and blocked with the blocking solution supplied in the Duolink in situPLA kit (Sigma-Aldrich).” is incomplete and should be expanded as “For assaying the associationsof eHSP90awith CD91 and TLR4, PBS or rHSP90a-treated macrophages werefixed with 3% paraformaldehyde and then blocked with the blocking solution supplied in the Duolink in situPLA kit (Sigma-Aldrich). For assaying CD91−MyD88, TLR4−MyD88, CD91−IRAK1, MyD88−IRAK1, MyD88−JAK2, and MyD88−TYK2 associations, PBS or rHSP90a-treated macrophages werefixed with 3% paraformaldehyde, permeabilized with 0.1% Triton X-100, and then blocked with the blocking solution…” (lines 202-208 of the second revised manuscript).